# Evaluation of Antibiotic Prescribing Pattern Using WHO Access, Watch and Reserve Classification in Kinshasa, Democratic Republic of Congo

**DOI:** 10.3390/antibiotics12081239

**Published:** 2023-07-27

**Authors:** Jocelyn Mankulu Kakumba, Jérémie Mbinze Kindenge, Paulin Mutwale Kapepula, Jean-Marie Liesse Iyamba, Murielle Longokolo Mashi, Jose Wambale Mulwahali, Didi Mana Kialengila

**Affiliations:** 1Laboratory of Drug Analysis, Faculty of Pharmaceutical Sciences, University of Kinshasa, Kinshasa XI 212, Democratic Republic of Congo; jeremiembinze@gmail.com (J.M.K.); didi.mana@unikin.ac.cd (D.M.K.); 2Centre d’Etudes des Substances Naturelles d’Origine Végétale (CESNOV), Faculty of Pharmaceutical Sciences, University of Kinshasa, Kinshasa XI 212, Democratic Republic of Congo; paulin.mutwale@unikin.ac.cd; 3Laboratory of Experimental and Pharmaceutical Microbiology, University of Kinshasa, Kinshasa XI 212, Democratic Republic of Congo; liesse.iyamba@unikin.ac.cd (J.-M.L.I.); josewambale@gmail.com (J.W.M.); 4Département de Médecine Interne, Service de Maladies Infectieuses et Tropicales, Kinshasa XI 212, Democratic Republic of Congo; muriellelongokolo@gmail.com

**Keywords:** antibiotic, prescribing pattern, prescription, antimicrobial resistance, WHO AWaRe classification

## Abstract

Background: The AWaRe tool was set up by the World Health Organization (WHO) to promote the rational use of antimicrobials. Indeed, this tool classifies antibiotics into four groups: access, watch, reserve and not-recommended antibiotics. In The Democratic Republic of Congo, data on antibiotic dispensing (prescribing) by health professionals according to the AWaRe classification are scarce. In this research work, we aimed to explore antibiotic dispensing pattern from health professionals according to the WHO AWaRe classification to strengthen the national antimicrobial resistance plan. Methods: For this purpose, a survey was conducted from July to December 2022 in the district of Tshangu in Kinshasa. From randomly selected drugstores, drug-sellers were interviewed and randomly selected customers attending those drugstores were included in the study for medical prescriptions collection. The prescribed antibiotics were classified into the access, watch, reserve and not-recommended antibiotics group and by antibiotics number by prescription among pharmacies surveyed. Results: 400 medical prescriptions were collected from 80 drugstores and among which, 301 (75.25%) contained antibiotics. Out of 301 prescriptions, we noticed 164 (54.5%) containing one antibiotic, 117 (38.9%) containing two antibiotics, 15 (5%) containing three antibiotics and 5 (1.6%) containing four antibiotics. A total of 463 antibiotics were prescribed and distributed as 169 (36.5%) were from the access group, 200 (43.2%) from the watch group and 94 (20.3%) from not-recommended antibiotics group, respectively. This can explain the fact of emerging bacterial strains, as, according to the WHO recommendations, the access group should be prioritized because of its activity against a wide range of commonly encountered pathogens and its showing low resistance susceptibility compared to antibiotics from other groups. Based on the anatomical, therapeutic and chemical (ATC) classifications, we observed that third generation cephalosporins represented 34.33% of the prescribed antibiotics, followed by penicillins (17.17%), macrolides (7.63%), aminoglycosides (7.36%) and Imidazole (7.36%), thus accounting approximately for 74% of the classes of antibiotics prescribed. Additionally, among them, the most frequently prescribed antibiotics were Ceftriaxone (21.38%), Amoxicillin (11.01%), Gentamycin (5.61%), Amoxicillin-clavulanic acid (5.61%), Azithromycin (4.97%) and Metronidazole (4.75%), thus accounting for approximately 54% of all the prescribed antibiotics. Conclusion: These results highlight the importance of strict implementation of the national plan to combat antimicrobial resistance and the need to train health workers in the correct application of the WHO AWaRe classification.

## 1. Introduction

Antimicrobials play a major role in the fight against infectious diseases around the world. Unfortunately, excessive and abusive use of antimicrobials is at the origin of the emergence and spread of resistance to a large number of antimicrobials, making them less and less effective against infectious diseases. Antimicrobial resistance (AMR) threatens the survival of all humanity and is, therefore, becoming a major public health problem worldwide [1,2]. 

Although AMR is a problem that affects all regions of the world, it remains more pronounced in Sub-Saharan African countries because projections made by the World Health Organization (WHO) show that if adequate provisions are not taken in time, the AMR would cause more than 4,150,000 deaths by 2050 [3]. AMR is also the root of several other negative consequences; for example, on the health level (inability of healthcare personnel to cure certain infectious diseases that were once easy treated, such as tuberculosis, gonorrhea, sepsis, etc.), economically (increased cost related to medical care) and socially (prolonged hospital stay and loss of patient confidence in the health care system).

Several factors promote the development of AMR. We can cite, for example, on one side, the irrational medical prescriptions of antibiotics and other antimicrobials in humans, self-medication, inappropriate veterinary prescriptions in animals, the abusive use of antimicrobial on crops, as well as the dissemination of residues of these antibiotics in the environment [4,5,6] and, on the other side, the lack of: application of antibiotics use guidelines, modern diagnostic tools, motivation and the insufficiency of qualified health personnel for laboratory diagnosis, lack of access to clean water, sanitation, and hygiene for humans and animals; poor infection prevention and control measures in hospitals; poor access to medicines and vaccines; lack of awareness and knowledge; and irregularities with legislation, etc. [5,6].

Antibiotic resistance is an integral part of AMR; however, it is considered the most dangerous and urgent global risk that requires more than special attention from all health actors and political authorities worldwide [1,2]. AMR is a critical global problem affecting humans, the environment, and animals and requires one health approach to combat it [1].

In Africa, in general, and in the Democratic Republic of Congo (DRC), in particular, a policy for monitoring antibiotic resistance and the rational use of antibiotics is still lacking. The absence of an adequate quality assurance system in the pharmaceutical sector facilitates the excessive and irrational use of substandard antibiotics in hospitals and in the population, in animal husbandry and in agriculture, which leads to the rapid selection of multi-resistant strains [5,6,7].

Research work on antibiotic resistance initiated in the DRC showed a strong trend towards an increase in the rate of resistance to commonly indicated antibiotics by strains of *Staphylococcus aureus* and extended-spectrum beta-lactamase-producing Enterobacteriaceae [8,9,10,11].

In its National Health Development Plan (NHDP) 2016–2020, the DRC set itself the general objective of contributing to the improvement of the health status of the population, in order to enable everyone to live in good health and promote well-being for all at all ages within the framework of universal health coverage by 2030 [12]. Nevertheless, this increasing trend towards the resistance to antibiotics observed in the DRC mitigates the achievement of this noble goal.

To deal with this phenomenon, more and more encouraging initiatives are being taken. Thus, in collaboration with the Ministry of Public Health and Hygiene, it was developed in support of the Project “Surveillance of antimicrobial resistance in the DRC, a University Reference Center for Research and Surveillance of Antimicrobial Resistance” (CURS-RAM), the missions of which are training and research on AMR, surveillance and service to society. Another project, the Interuniversity Innovation Center for the Implementation of a Green Approach to Combating Antimicrobial Resistance (PI-RAM), the mission of which is to develop technological innovations capable of combating AMR by green approach based on nanotechnology, works in collaboration with the CURS-RAM within the University of Kinshasa in the Faculty of Pharmaceutical Sciences.

The global action plan to combat AMR aims to enable humanity to continue to have, for as long as possible, effective means of preventing and treating infectious diseases in the form of safe and effective drugs, guaranteed quality, used responsibly and accessible to all who need it [1,13]. Based on the global action plan to combat AMR, the DRC developed its national plan to combat AMR which has set itself, among other objectives, to optimize the use of antimicrobial drugs in human health, animal and plant by promoting the rational use of antimicrobials.

The WHO defines the rational use of medicines as the prescription of the most appropriate product, obtained in time and at an affordable price for all, delivered correctly and administered at the appropriate dosage and for an appropriate period of time [14]. This definition clearly shows that all the actors (doctor, pharmacist, patient or his relatives) intervening in the circuit of dispensing or consumption of drugs must play their roles correctly. However, the global health situation in low incomes countries in general, and in the DRC in particular, does not favor the rational use of drugs. Indeed, globally, more than 50% of all medicines are prescribed, dispensed or sold inappropriately [15]. In DRC, several studies reported the misuse of drugs in general and antibiotics in particular [16,17,18,19].

The WHO developed the access, watch, reserve (AWaRe) classification of antibiotics with the aim of combating the irrational use of antibiotics and curbing the growing resistance to antimicrobials. The AWaRe tool classifies antibiotics into four groups and specifies which antibiotics to use for the most common and serious infections, which should be always available in the healthcare system (antibiotics whose accessibility is essential) and which should be used in moderation or stored (antibiotics to be used selectively) and those used only as a last resort (reserve antibiotics). The fourth includes combinations of antibiotics whose indications and efficacy were not proven [20].

The primary objective of this classification was to bring to at least 60% of the proportion of the world’s consumption of antibiotics in the group of antibiotics whose accessibility is essential and to reduce the use of the antibiotics most exposed to the risk of resistance in the groups of antibiotics to be used selectively and reserved antibiotics. The use of antibiotics whose accessibility is essential reduces the risk of resistance because they are narrow-spectrum antibiotics. They also cost less because they are available in generic formulations [20].

This tool represents hope for humanity in the fight against AMR, it aroused great interest among scientists, which explains the growing number of articles written to evaluate its use [21,22,23,24,25]. Unfortunately, all these studies showed that there is still a long way to go to reach an appreciable application of the AWaRe tool.

In low-income countries such as the DRC, where any medicine, or even antibiotics, can be obtained without a medical prescription, the problem of the emergence of AMR cases is a permanent concern. Among them, we can mention the use of Amoxicillin (from penicillin family) and Tetracycline (Cyclines family) as part of the most used antibiotics without prescription. The public indulges in the misuse of antibiotics by practicing self-medication and health personnel over-prescribing antibiotics following the non-application of laws governing the pharmaceutical sector [19].

To the best of our knowledge, no study was yet conducted on antibiotics prescribing pattern in the district of Tshangu, in the city of Kinshasa, according to the WHO AWaRe tool. Tshangu, as the largest district of the city, is located in the eastern part and gathers the largest popular mass (approximately 17.07 millions) in their multiculturalism. Additionally, it was subjected into this research because it is full of the majority of illegal pharmaceutical and medical establishments where illegal prescriptions and dispensing of pharmaceuticals are common. Additionally, under such conditions, we are almost reassured that the implementation of the WHO prescribing standards for antibiotics is bound to encounter enormous difficulties. It is in this context that this study to explore the adherence of prescribers to the WHO protocol on the rational use of antibiotics according to WHO AWaRe tool was initiated.

## 2. Materials and Methods

### 2.1. Study Environment

The capital of the DRC, the city–province of Kinshasa, is made up of four districts, namely: Funa, Lukunga, Mont Amba and Tshangu. The district of Tshangu is located east of Kinshasa, it is by far the most extensive district of Kinshasa, but also the most rural (4°21′0″ S et 15°22′60″ E). There is a promiscuity ((high popular density with large number of pharmacies) that does not facilitate the application of laws related to the regulation of the health system, thus illustrating an ideal framework for studying the quality of care administered to the population. Targeted pharmacies (of interest) were randomly selected based on the availability of their drug-sellers to the conduct of this study. Those pharmacies were selected in the field due to their frequentation by patients. No geographic limit was set, as most of them are concentrated in areas of high population density. However, it should be pointed out that some of the 80 pharmacies we also worked on were isolated in certain parts of the district where there were few pharmacies open to the public.

### 2.2. Administrative Documents 

Authorization to undertake the study: first of all, the authorization to undertake this study was requested and obtained from the National Ethics Commission.

Medical prescriptions: a photocopy of medical prescriptions was obtained from patients who gave their consent to participate in the study.

A voluntary consent document, ensuring the anonymity of the various participants who agreed to participate in the study, was signed.

Data collection form: a form was established for data collection following the WHO model [14].

AWaRe list: document drawn up by WHO containing the classification of antibiotics into groups, access, watch, reserve and not-recommended antibiotics group.

### 2.3. Methods Used

A retrospective and descriptive study was conducted from July to December 2022 in the district of Tshangu, with the aim of evaluating the adherence of prescribers to the WHO protocol on the rational use of antibiotics according to the WHO’s AWaRe tool.

### 2.4. Data Collection and Analysis

A total of 400 prescriptions were randomly collected to constitute the sample used in this study. Only prescriptions from the Tshangu district were retained, illegible prescriptions were excluded from this study. One out of three patients who came to obtain medication was approached to request their participation in the study.

The Statistical Package for Social Sciences program version 20.0 (SPSS) from the University of Kinshasa was used to perform the statistical analyses.

Frequencies and percentages were used to express the results.

The models used were described according to the WHO 2019 AWaRe classification of antibiotics, in which antibiotics were classified into four groups: access, watch, reserve and not-recommended combinations [26]. For medical prescriptions containing more than one antibiotic, their classification would be made by giving priority to the highly restricted antibiotic. For example, if the medical prescription contained both access group and watch group antibiotics, the prescription will be classified as containing the watch group antibiotics. We also used the anatomical, therapeutic and chemical classification, developed by the WHO to prioritize the consumption of pharmacological classes of antibiotics.

## 3. Results

This study revealed an overuse of antibiotics. Indeed, of the 400 prescriptions collected, 301 (75.25%) prescriptions contained antibiotics, 164 (164/301; 54.5%) contained one antibiotic, 117 (117/301; 38.9%) contained two antibiotics, 15 (15/301; 5%) contained three antibiotics and 5 (5/301; 1.6%) contained four antibiotics. These collections were carried out from 80 randomly selected pharmacies according to their frequentation frequency and the drug selling price. Comprising a large population, the majority of whom are poor, the district of Tshangu is characterized by the presence of pharmacies that are often unauthorized (not having official opening permits), so that for most of them, this is a lucrative business activity. Thus, the only concern of these shopkeepers is to sell and not to provide quality care.

In the group of prescriptions containing one antibiotic, Ceftriaxone injectable, even if belonging to the watch category, was the most prescribed drug, accounting for 31.1% (51/164) of prescriptions containing an antibiotic, followed by Amoxicillin (access) 13.4% (22/164), Amoxicillin–Clavulanic acid combination (access) 12.2% (20/164), Azithromycin (watch) 10.4% (17/164) and Ciprofloxacin (watch) 7.4% (12/164). Of the 164 prescriptions containing one antibiotic, 54 (54/164; 32.9%) prescriptions contained antibiotics of access group and 110 (110/164; 67.1%) of watch group. These results are shown in Table 1.

Of the 117 prescriptions including two antibiotics per prescription, Ceftriaxone (watch) was once again the most prescribed antibiotic. Indeed, it was present on 43 (43/117; 36.7%) prescriptions, followed by Amoxicillin (access) which was present on 27 (27/117; 23.08%), Gentamicin (access) which was present on 23 (23/117; 19.7%) and Metronidazole (access) which was present on 16 (16/117; 13.67%). A total of 21 (21/117; 18%) prescriptions contained access category antibiotics and 62 (62/117; 53%) prescriptions contained watch category antibiotics, of which 54 (54/117; 46.15%) were composed of one antibiotic from the access group and another from the watch group and 8 (8/117; 6.84%) of two antibiotics from the watch group and 34 (34/117; 29%) prescriptions contained fixed-dose combinations antibiotics in the not-recommended group. These results described above are presented in Table 2.

Of the 15 prescriptions containing three antibiotics each, 1 (1/15; 6.67%) prescription was from the access group, 7 (7/15; 46.7%) prescriptions from the watch group, of which 4 (4/15; 26.67%) consisted of one antibiotic from the watch group and two from the access group and 3 (3/15; 20%) of two antibiotics from the watch group and one access group, and 7 (7/15; 46.67% ) prescriptions containing not-recommended fixed-dose combinations; of which 3 (3/15; 20%) were associated with an access group antibiotic and 4 (4/15; 26.27%) with a watch group antibiotic (Table 3).

In the 5 prescriptions comprising four antibiotics, it can be noted that all 5 (100%) contained the not recommended fixed-dose combinations. All being associated with one antibiotic from the Access group and another from the Watch group (Table 4).

In addition to the three main groups of the AWaRe list, this list includes a fourth group of antibiotics not recommended by the WHO due to their unproven indication or efficacy. In this study, a total of 46 (46/301; 15.28%) prescriptions contained antibiotics of the not-recommended group. The most prescribed combinations were Ceftriaxone/Sulbatam (13/46; 28.26%), Norfloxacin/Metronidazole (10/46; 21.74%), Ofloxacin/Ornidazole (9/46; 19.56%) and Ciprofloxacin/Metronidazole (8/46; 17.39%). There was even one (1/46; 2.17) prescription containing together two fixed-dose combinations of antibiotics that were not recommended: Ceftriaxone/Sulbactam and Norfloxacin/Metronidazole (Table 4).

In Table 5, a summary table that shows all the antibiotics prescribed out of the 301 prescriptions, we can notice that: Ceftriaxone (watch) was the most prescribed drug (21.38%) (101/463) of the antibiotics prescribed, followed by Amoxicillin (access) (11.01%) (51/463), Amoxicillin–Clavulanic acid combination (access) (5.61%) (26/463), Gentamicin (5.61%) (26/463), Azithromycin (watch) (4.97%) (23/463) and Metronidazole (access) (4.75%) (22/463). Of the 463 antibiotics listed in this study, 169 (169/463; 36.50%) belonged to the access group, 200 (200/463; 43.20%) to the watch group and 94 (94/463; 20.3%) to the group of not-recommended antibiotics.

Applying the ATC classification to the 463 antibiotics prescribed, it can be seen that 128 (128/463; 27.65%) antibiotics were third generation cephalosporins (watch), followed by 63 (63/463 13.61%) which were Penicillins (access), 28 (28/463; 6.05%) which belonged to Macrolides (watch), 27 (27/463; 5.83%) which were Imidazoles (access), 27 (27/463; 5.83%) Aminoglycosides (access), 26 (26/463; 5.61%) Beta-lactamase inhibitors and 22 (22/463; 4.75%) which were Fluoroquinilones (watch). A total of 169 (169/463; 36.50%) antibiotics belonged to the access category, 200 (200/463; 43.20%) to the watch category and 94 (94/463; 20.3) to the group of antibiotics not recommended (see Table 6).

## 4. Discussion

As mentioned above, our study was conducted under authorization of the National Ethics Commission, taking into account the promiscuity resulting from high number of pharmacies in this district that do not comply with the limits set by the country’s pharmaceutical legislation authority, which recommends a distance of 500 meters in urban areas and around 1 kilometer in rural areas. Additionally, this leads to their concentration where there are high population movements. within the Tshangu population, number of unauthorized pharmacies and the lack of respect to the antibiotics prescriptions guidelines established by the WHO. This study demonstrated that antibiotics were over-used in the district of Tshangu because most of them are prescribed in a framework that does not take into account the characteristics of the infections involved, or even for the simple reason of suspicion of an unconfirmed infection. Additionally, above all, the majority of them are prescribed to patients without prior establishment of an antibiogram in accordance with the general rules. Indeed, the collected data showed that out of 400 prescriptions, 301 (301/400; 75.25%) contained antibiotics, highlighting the frequency of use of these products compared to others for other therapeutic indications. Additionally, out of the 301 prescriptions containing antibiotics, 76 (76/301; 25.25%) prescriptions contained antibiotics belonging to the access category, 179 (179/301; 59.47%) those of the watch category and 46 (46/301; 15.28%) of the not-recommended category. By developing the AWaRe tool, the WHO’s objective was to encourage the use of access group antibiotics, up to 60%, because these are antibiotics that target a limited number of germs and, therefore, would not easily favor AMR. However, in this study, we noted a lower use of antibiotics from the access group, i.e., 25.25%.

Of the 301 prescriptions with antibiotics, 164 (164/301; 54.5%) medical prescriptions contained one antibiotic, 117 (117/301; 38.9%) contained two antibiotics, 15 (15/301; 5%) contained three antibiotics and 5 (5/301; 1.6%) contained four antibiotics.

### 4.1. Single Antibiotic Prescription

A total of 54.5% (164/301) of prescriptions had one antibiotic; this percentage is much lower than that found by Md. Ariful Islam et al. in Bangladesh, who noted 87.3% medical prescriptions containing one antibiotic [23]. This may be explained by the fact that in the DRC, due to the lack of appropriate medical examinations, prescribers tend to prescribe more than one antibiotic to broaden the spectrum of action, despite the obvious danger of AMR risk.

Of these 164 antibiotics prescribed, 32.9% (54/164) were from the access group while 67.1% (110/164) were from the watch group. We noted that prescriptions of watch group dominated, i.e., 67.1%. This result is similar to that found in a study conducted in West Asia (66.1%) [24] and Kazakhstan (68%) [22], the result is higher than that of a study conducted in Bangladesh (53.6%) [23] and is lower than the result found in Armenia (87.9%) and Jordan (84.4%) [24]. Indeed, watch group antibiotics should not be used in large quantities, as noted here, because they have broad action spectra and are, therefore, likely to promote the selection of resistant germs. This tendency to misuse watch group antibiotics is characteristic of low-income countries (DRC, Armenia, Jordan and Kazakhstan). This may be explained by the facts that in these low-income countries, the irrational use of antibiotics is common, which suggests the existence of resistance to access group antibiotics [8,9,10,11,27].

Of the 19 different antibiotics prescribed out of these 164 prescriptions, 5 antibiotics accounted for 74.5%. They were distributed as follows: Ceftriaxone (watch) was the most frequently prescribed 31.1% (51/164), followed by Amoxicillin (access) 13.4% (22/164), Amoxicillin–Clavulanic Acid (access) 12.2% (20/164), Azithromycin (watch) 10.4 (17/164) and Ciprofloxacin (watch) 7.4% (12/164). The watch group accounted for 48.9% and the access group only 25.6% of the top five. A similar finding on Ceftriaxone was noted in a study which focused on the prescription of antibiotics in the hospital sector in adults; the study concerned 69 countries, but with a slightly lower percentage of 24.8%. This, unfortunately, shows an overuse of watch group antibiotic on a global level, with a serious risk of favoring AMR towards Ceftriaxone over time.

### 4.2. Prescription of Two Antibiotics

A total of 117 (117/301; 38.9%) prescriptions contained two antibiotics. This percentage is much higher than the 11.2% found in Bangladesh [23]. Indeed, the tendency to combine antibiotics is very risky because it promotes the development of AMR. Among the most used antibiotics, we can cite Ceftriaxone (watch) which accounted for 36.75% of prescriptions, followed by Amoxicillin (access) (23.08%) and Gentamicin (access) (19.7%). These results are much higher than those found in Bangladesh, Ceftriaxone 0.7% of prescriptions, Amoxicillin (access) (5.5%) and Gentamycin (access) (0.9%) [23]. Our study showed a high tendency to prescribe a limited number of antibiotics, particularly those of the watch group, thus increasing the risk of developing AMR.

Indeed, because of the lack of appropriate medical equipment (appropriate clinical laboratory) to carry out clinical examinations, prescribers tend to prescribe the same antibiotics already used, of which they are sure of the results. In addition, 21 prescriptions (17.94%) of prescriptions contained access category antibiotics, 62 (62/117; 53%) prescriptions contained watch category antibiotics and 34 (34/117; 29.06%) prescriptions contained not-recommended antibiotics. Indeed, the 29.06% of prescriptions containing not-recommended antibiotics could be explained by the fact that these combinations of antibiotics are considered to be the antibiotics of choice and are legal to use in the DRC while not recommended by the WHO. The AWaRe tool was set up to increase the use of access group antibiotics beyond 60%; the result found in our study (17.94%) showed that health authorities still have a lot to do to encourage prescribers to promote the rational use of antibiotics.

### 4.3. Prescription of Three Antibiotics

A total of 15 (15/301; 5%) prescriptions contained three antibiotics. The Bangladesh study found 1.4% of medical prescriptions containing three antibiotics, a much lower percentage than the 5% found in our study. Poly-pharmacy is a dangerous practice for drugs in general and antibiotics in particular because the risk of drug interactions and AMR is high. One (6.67%) prescription included antibiotics from the access category, seven (7/15; 46.7%) from the watch category, which is a lower percentage than the 87.9% found in Armenia and 84.4% found in Jordan [24], and seven (7/15; 46.7) from antibiotics of the not-recommended group.

### 4.4. Prescription to Four Antibiotics

A total of five (5/301; 1.67%) prescriptions contained four antibiotics each. The tendency to poly-pharmacy was even more accentuated in this case. With prescriptions containing four antibiotics, all combinations belonged to the not-recommended group, which greatly increases the risk of AMR and dangerous drug interactions for the patient.

### 4.5. Summary Study of 301 Medical Prescriptions

A total of 463 antibiotics were prescribed out of the 301 medical prescriptions. The most prescribed antibiotics were Ceftriaxone (watch) representing 21.81% (101/463), followed by Amoxicillin (access) representing 11.01% (51/463), the Amoxicillin–Clavulanic Acid combination (access) representing 5.61% (26/463), Gentamicin representing 5.61% (26/463), Azithromycin (watch) representing 4.97% (23/463), Metronidazole (access) representing 4.75% (22/463) and Ciprofloxacin representing 3.67% (17/463). A similar study was conducted by Yingfen Hsia et al, which covered 56 countries from different continents and the following results (Table 7) were found in relation to the use of antibiotics: 

The percentages of consumption for all these antibiotics were higher in our study than those found in these four continents except for amoxicillin–clavulanic acid in Europe (9.1%) and Southwest Asia (8.9 %), Gentamycin (10%) and Ciprofloxacin (3.8%) in Africa and finally Metronidazole 5.5% and 5% in Africa and Asia, respectively. Once again, Ceftriaxone (watch group) was the most frequently prescribed antibiotic. Ceftriaxone misuse is higher not only in DRC but also around the world, and so, appropriate decisions must be taken to avoid AMR by promoting Ceftriaxone rational use.

Out of 463 antibiotics prescribed, 169 (169/463; 36.5%), 200 (200/463; 42.3%) and 94 (94/463; 20.2%) were from the access, watch and not-recommended antibiotics groups, respectively. These results clearly show that the WHO objective of achieving 60% use of access group antibiotics in the DRC is still an ideal [20]. An amount of 36.5% of access group antibiotics found in our study was lower than 59% found in Vietnam [21], but it was similar to the 36.4% found in Bangladesh [23]. This low use of access group antibiotics is characteristic of poor countries where the lack of modern equipment, of information on the correct use of antibiotics and the non-application of national policies to combat AMR causes antibiotics misuse. These results showed a total absence of reserve group antibiotics. This result is identical to that found by Nam Vinh Nguyen et al. [21] and lower than the 10% found by Md. Ariful Islam [23]. The absence of reserve group antibiotics is explained by the fact that in the DRC, these antibiotics are not available in pharmacies open to the public. 

The AWaRe list includes a list of therapeutic fixed-dose combinations of a number of broad-spectrum antibiotics with unproven efficacy. They are, therefore, not recommended by the WHO for use in clinical practice [28]. A total of 20.3% (94/463) of the antibiotics prescribed in this study were from the group of not-recommended antibiotics. This result is much higher than the 1.7% found in Vietnam. This high percentage of antibiotics in the not-recommended antibiotics group shows how the irrational use of antibiotics reached a worrying scale. Indeed, many of these combinations are legal tender in the DRC. Thus, it can be easily obtained in pharmacies and often even without medical prescriptions. The most prescribed not recommended fixed-dose therapeutic combinations were Ceftriaxone–Sulbatam (5.61%) (28/463), Norfloxacin–Metronidazole (4.32%) (20/463), Ofloxacin–Ornidazole (3.89%) (18/463), and Ciprofloxacin–Metronidazole (3.45%) (16/463) of all antibiotics prescribed. The irrational use of these therapeutic combinations of antibiotics, which are not recommended in the reference documents due to their unproven efficacy, poses a proven risk of selection of resistant microorganisms.

A total of 42.30% of antibiotics used in prescriptions were from the watch group, including oral forms of third generation Cephalosporins (Cefixime and Cefpodoxime), Fluoroquinolones (Levofloxacin and Ciprofloxacin) and macrolides (Azithromycin). As well as their negative effect on the intestinal flora, these broad-spectrum antibiotics facilitate the development of antimicrobial resistance [29,30,31,32].

Several strengths characterized this study. First, to the best of our knowledge, this is the first study to assess the antibiotic prescription pattern of health personnel in the district of Tshangu in the city of Kinshasa. Second, we established the profile of medical prescriptions containing one, two, three and four antibiotics, which demonstrated excessive use of watch group antibiotics and not-recommended antibiotics. Third, an overall assessment of all prescriptions according to AWaRe classification and the anatomical, therapeutic and chemical (ATC) classification was performed. The latter showed the overuse of certain therapeutic families (third generation cephalosporins, macrolides, beta-lactamase inhibitors, etc.) which are likely to promote AMR.

However, some limitations should be considered when interpreting these results. Indeed, our work was limited only to the district of Tshangu, which is only one of four districts of the city of Kinshasa. In addition, only a few pharmacies in Tshangu district participated in the study, thereby providing a limited sample of 400 medical prescriptions. Finally, the study did not cover the whole year, thus not considering the possible influences of seasonal variation on the qualities of medical prescriptions.

## 5. Conclusions

We found an irrational and exaggerated use of watch group antibiotics and non-recommended antibiotics. Certain classes of antibiotics such as third generation Cephalosporins, Penicillins, Macrolides, Aminoglycosides and Imidazoles alone constituted approximately 74% of the classes of antibiotics used. In addition to this, Ceftriaxone was the most prescribed antibiotic, accounting for 21.38% of the antibiotics prescribed. This finding underlines the need to popularize the use of the AWaRe tool among health personnel to encourage responsible use of antibiotics and, thus, combat the emergence of AMR. Unfortunately, our study was limited to a survey and collection of prescriptions arriving in pharmacies without knowing the problems for which they were prescribed exactly. As a tool to support antibiotic stewardship efforts at local, national and global levels, AWaRe classifies antibiotics into three groups (access, watch and reserve), taking into account the impact of different antibiotics and their classes on antimicrobial resistance, to expand their rational use impact. In addition to DRC having a national plan to fight against antimicrobial resistance, it is time that the authorities with public health in their prerogatives can ensure the strict implementation of these objectives and the application of laws governing the health sector to limit the excessive use of antibiotics.

## Figures and Tables

**Table 1 antibiotics-12-01239-t001:** AWaRe Classification of antibiotics (prescriptions with one prescribed antibiotic).

No.	Antibiotics Generic Names	n = 164/(%)	Category (%)
1	Amoxicillin	22 (13.4)	Access 54:32.9%
2	Amoxicillin/clavulanic-acid	20 (12.2)	
3	Flucloxacillin	5 (3,0)	
4	Cefadroxil	2 (1.2)	
5	Metronidazole	2 (1.2)	
6	Doxycycline	1 (0.6)	
7	Penicillin	1 (0.6)	
8	Cefalexin	1 (0.6)	
9	Ceftriaxone	51 (31.1)	Watch 110:67.1%
10	Azithromycin	17 (10.4)	
11	Ciprofloxacin	12 (7.3)	
12	Cefixime	10 (6.1)	
13	Lincomycin	6 (3.7)	
14	Cefotaxime	4 (2.4)	
15	Cefpodoxime	3 (1.8)	
16	Erythromycin	3 (1.8)	
17	Ofloxacin	2 (1.2)	
18	Levofloxacin	1 (0.6)	
19	Streptomycin	1 (0.6)	
	TOTAL	164 (100)	164:100%

**Table 2 antibiotics-12-01239-t002:** AWaRe Classification of antibiotics (prescriptions with two prescribed antibiotics).

No.	Antibiotics Generic Names	n = 117/(%)	Category (%)
1	Amoxicillin + Gentamicin	6 (5.13)	Access 21 (18)
2	Amoxicillin + Furazidin	1 (0.85)	
3	Amoxicillin + Amikacin	1 (0.85)	
4	Amoxicillin/clavulanic-acid +Metronidazole	3 (2.56)	
5	Amoxicillin/clavulanic-acid +Furazidin	2 (1.71)	
6	Amoxicillin/clavulanic-acid + Ornidazole	1 (0.85)	
7	Metronidazole + Furazidin	2 (1.71)	
8	Metronidazole + Doxicycline	1 (0.85)	
9	Metronidazole + Cefadroxil	1 (0.85)	
10	Chloramphénicol + Gentamicin	1 (0.85)	
11	Flucloxacillin + Metronidazole	1 (0.85)	
12	Penicillin + Doxycycline	1 (0.85)	
13	Ceftriaxone + Gentamicin	15 (12.82)	Watch 62 (53)
14	Ceftriaxone + Amoxicillin	11 (9.40)	
15	Ceftriaxone + Metronidazole	8 (6.84)	
16	Ceftriaxone + Lincomycin	3 (2.56)	
17	Ceftriaxone + Cefixime	1 (0.85)	
18	Ceftriaxone + Doxycycline	1 (0.85)	
19	Ceftriaxone + Ornidazole	1 (0.85)	
20	Ceftriaxone + Chloramphenicol	1 (0.85)	
21	Ceftriaxone + Clindamycin	1 (0.85)	
22	Ceftriaxone + Ciprofloxacin	1 (0.85)	
23	Lincomycin + Amoxicillin	2 (1.71)	
24	Lincomycin + Gentamicin	1 (0.85)	
25	Lincomycin + Kanamycin	1 (0.85)	
26	Lincomycin + Cefuroxime	1 (0.85)	
27	Lincomycin + Ofloxacin	1 (0.85)	
28	Azithromycin + Amoxicillin	2 (1.71)	
29	Azithromycin + Doxycycline	2 (1.71)	
30	Ciprofloxacin + Amoxicillin	1 (0.85)	
31	Ciprofloxacin + Cefadroxil	1 (0.85)	
32	Ciprofloxacin + Furazidin	1 (0.85)	
33	Cefotaxime + Chloramphenicol	1 (0.85)	
34	Cefotaxime + Amoxicillin	2 (1.71)	
35	Cefixime + Amoxicillin	1 (0.85)	
36	Kanamycin + Ornidazole	1 (0.85)	
37	Erythromycin + Penicillin	1 (0.85)	
38	Norfloxacin/Metronidazole	8 (6.84)	not recommended 34 (29)
39	Ofloxacin/Ornidazole	8 (6.84)	
40	Ciprofloxacin/Metronidazole	7 (5.98)	
41	Ceftriaxone/Sulbactam	6 (5.13)	
42	Ceftriaxone/Tazobactam	2 (1.71)	
43	Ciprofloxacine/Ornidazole	2 (1.71)	
44	Cefixime/Ornidazole	1 (0.85)	
TOTAL		117 (100)	117 (100)

**Table 3 antibiotics-12-01239-t003:** AWaRe classification of antibiotics (prescriptions with three prescribed antibiotics).

No.	Antibiotics Generic Names	n = 15/(%)	Category (%)
1	Penicillin + Chloramphenicol + Doxycycline	1 (6.67)	Access 1 (6.6)
2	Ceftriaxone + Gentamicin +Chloramphenicol	1 (6.67)	Watch 7 (46.7)
3	Ceftriaxone + Penicillin + Amoxicillin	1 (6.67)	
4	Ceftriaxone + Gentamicin + Furazidin	1 (6.67)	
5	Cefotaxime + Metronidazole + Doxycycline	1 (6.67)	
6	Ceftriaxone + Lincomycin + Gentamicin	1 (6.67)	
7	Lincomycin + Ceftriaxone + Sulbactam	1 (6.67)	
8	Cefotaxime + Azythromycin + Oxacillin	1 (6.67)	
9	Ceftriaxone + Sulbactam + Ornidazole	1 (6.67)	not recommended 7 (46.7)
10	Ceftriaxone + Sulbactam + Amoxicillin	1 (6.67)	
11	Ceftriaxone + Sulbactam + Metronidazole	1 (6.67)	
12	Ceftriaxone + Sulbactam + Cefixime	1 (6.67)	
13	Ceftriaxone + Sulbactam + Lévofloxacine	1 (6.67)	
14	Ciprofloxacin/Metronidazole + Cefotaxime	1 (6.67)	
15	Ofloxacin/Ornidazole (A) + Azythromycin	1 (6.67)	
TOTAL		15 (100)	15 (100)

**Table 4 antibiotics-12-01239-t004:** AWaRe classification of antibiotics (prescriptions with four prescribed antibiotics).

No.	Antibiotics Generic Names	n = 5/(%)	Category n (%)
1	Ceftriaxone/Metronidazole + Ofloxacin + Ornidazole	1 (20)	not recommended 5 (100)
2	Ceftriaxone/Sulbactam + Norfloxacin + Metronidazole	1 (20)	
3	Norfloxacin/Metronidazole + Erythromycin + Ornidazole	1 (20)	
4	Ceftriaxone/Sulbactam + Lincomycin + Metronidazole	1 (20)	
5	Ceftriaxone/Cefixime + Norfloxacin/Metronidazole	1 (20)	
TOTAL		5 (100)	5 (100)

**Table 5 antibiotics-12-01239-t005:** Antibiotics prescribed in Tshangu according to the WHO AWaRe classification.

No.	Antibiotics Generic Names	n = 463(%)	Category (%)
1	Amoxicillin	51 (11.01)	Access 169 (36.5)
2	Metronidazole	22 (4.75)	
3	Gentamicin	26 (5.61)	
4	Amoxicillin/acide clavulanic-acid	26 (5.61)	
5	Ornidazole	5 (1.08)	
6	Doxycycline	8 (1.73)	
7	Furazidin	7 (1.51)	
8	Flucloxacillin	6 (1.30)	
9	Chloramphenicol	5 (1.08)	
10	Penicillin	5 (1.08)	
11	Cefadroxil	4 (0.86)	
16	Cefalexime	1 (0.21)	
17	Clindamycin	1 (0.21)	
18	Oxacilline	1 (0.21)	
19	Amikacin	1 (0.21)	
20	Ceftriaxone	101 (21.38)	Watch 200 (43.20)
21	Ciprofloxacin	17 (3.67)	
22	Azithromycin	23 (4.97)	
23	Lincomycin	18 (3.89)	
24	Cefixime	14 (3.02)	
25	Ofloxacin	3 (0.65)	
26	Cefotaxime	10 (2.15)	
27	Erythromycin	5 (1.08)	
28	Cefpodoxime	3 (0.65)	
29	Kanamycin	2 (0.43)	
30	Levofloxacin	2 (0.43)	
31	Cefuroxime	1 (0.21)	
32	Streptomycin	1 (0.21)	
33	Ceftriaxone/Sulbactam	28 (5.61)	94 (20.30) Not recommended
34	Norfloxacin/Metronidazole	20 (4.32)	
35	Ofloxacin/Ornidazole	18 (3.89)	
36	Ciprofloxacin/Metronidazole	16 (3.45)	
37	Ceftriaxone/Tazobactam	4 (0.86)	
38	Ciprofloxacin/Ornidazole	4 (0.86)	
39	Cefixime/Ornidazole	2 (0.43)	
40	Norfloxacin/Metronidazole	2 (0.43)	
TOTAL		463 (100)	

**Table 6 antibiotics-12-01239-t006:** Proportional prescription of the WHO ATC chemical classes of antibiotics.

No.	WHO ATC Group	Antibiotics Classes	n = 463(%)	Category (%)
1	J01CF05	Penicillins	63 (13.61)	Access 169 (36.5)
2	J01XD01	Imidazoles	27 (5.83)	
3	J01GB03	Aminoglycosides	27 (5.83)	
4	J01CR02	Beta-lactamase inhibitor	26 (5.61)	
5	J01AA02	Tetracyclines	8 (1.73)	
6	J01XE01	Nitrofuran-derivatives	7 (1.51)	
7	J01BA01	Amphenicols	5 (1.08)	
8	J01DB05	First-generation cephalosporins	5 (1.08)	
9	J01FF01	Lincosamides	1 (0.21))	
10	J01DD04	Third-generation cephalosporins	128 (27.65)	Watch 200 (43.2)
11	J01FA10	Macrolides	28 (6.05)	
12	J01MA12	Fluoroquinolones	22 (4.75)	
13	J01FF02	Lincosamides	18 (3.89)	
14	A07AA08	Aminoglycosides	3 (0.65)	
15	J01DC02	Second-generation cephalosporins	1 (0.21)	
16		Third-generation-cephalosporins/Beta-lactamase-inhibitors	30 (6.48)	Not-recommended 94 (20.3)
17		Fluoroquinolones/Imidazoles	60 (12.96)	
18		Third-generation-cephalosporins/Imidazoles	4 (0.86)	
TOTAL			463 (100)	

**Table 7 antibiotics-12-01239-t007:** Pooled results from a similar study by continent [25].

Antibiotics	Africa	America	Europe	Southwest Asia
Amoxicillin	6.4%	3.3%	3.6%	2.5%
Amoxicillin + Clav. Acid	4.9%	2.4%	9.1%	8.9%
Azithromycin	2.1%	3.2%%	3.4%	1.9%
Ceftriaxone	12.0%	9.8%	10.1%	15.4%
Ciprofloxacin	3.8%	2.1%	2.9%	2.5%
Gentamicin	10.0%	1.6%	4.3%	0.0%
Metronidazole	5.5%	4.7%	3.7%	5.0%

## Data Availability

All data related to this research are contained within this paper.

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
