# Peer review of "Evaluation of Antibiotic Prescribing Pattern Using WHO Access, Watch and Reserve Classification in Kinshasa, Democratic Republic of Congo"

_antibiotics, 2023, doi:10.3390/antibiotics12081239_

Round 1

Reviewer 1 Report

This paper addresses an important issue, and provides additional evidence on the patterns of use of antibiotics and the application of the WHO antibiotic classification. There are some areas of the methods section that need to be addressed and / or clarified, but with these clarifications, the paper would be suitable for publication.

Abstract

Although the methods refers to the WHO classification of antibiotics into Access, Watch and Reserve categories, the results as described in the abstract are quite detailed and could be reduced to focus on the distribution this classification. It would be useful to include the number of drugstores included in the survey.

Introduction:

The rationale in general in regards the importance of AMR is outlined – but could be strengthened with more information on the role / contribution of antibiotic prescribing patterns, and rationale for selection of the district of Tshangu. No information is provided on the characteristics of Tshangu in terms of SES of population, availability / access to medical care, or the number and types of providers of pharmaceuticals.

Line 136 notes that many medicines are obtained without prescription. A potential limitation of the study is that it focuses on prescriptions and prescribing of antibiotics, and does not capture the potential use of antibiotics without prescription. This could be mentioned.

Methods
It would be customary for the methods section to precede the results, so that the reader could understand how the results were obtained. Suggest that the methods and materials section at the end of the paper in the current draft be moved to this position.

Results

More information on the distribution of the 400 prescriptions would be useful. For example, the number of pharmacies / drug sellers involved (line 348 refers to ‘only a few pharmacies’) and the type of these pharmacies could be relevant to the sample. For example, relatively lower cost pharmacies might attract relatively poorer customers, who might attend the lower cost providers to obtain prescriptions, and thus receive poorer care.

Table 1: The term ‘prescribed one by prescription’ in the Title is not clear. Suggest Classification of antibiotics prescribed as the single agent in prescription.

Similarly, the titles for subsequent Tables 2 and 3 could be clarified as : Antibiotics prescribed in prescriptions of two antibiotic agents; and Table 3: Antibiotics prescribed in prescriptions of three antibiotic agents.

Line 194 Gentamycin – misspelt, should be gentamicin

Discussion

It would be useful to commence the discussion with some comments on the methodology and its impact on the results. In particular the selection of the pharmacies and prescriptions involved and the degree to which these are representative of the use of antibiotics in DRC / Kinshasa.

Line 209-210. The statement that antibiotics were over-used cannot be justified on the basis of information on the number of prescriptions. It would need to be compared with the extent and types  of infectious disease. It might be more appropriate to say that they constitute a large proportion of the prescriptions.

The discussion section tends to repeat the results while comparing to studies in other countries. It would be appropriate to consider the potential strengths and weaknesses of the study methodology, and factors that might contribute to the high use of watch and not recommended categories, for example the expectation of patients, and existing levels of antibiotic resistance among common pathogens.

Line 260 ‘medical examination tools’ – this could be further clarified for example ? potential for laboratory examination ? Tendency to rely on ‘clinical’ exam.

Line 266 ‘legal tender in DRC’ – unclear what this means

3.4 Summary

Tends to repeat the information in the results and compare with studies in other countries – but difficult to follow. A table with columns for each study might be easier to compare.

Some reference to the methodology at the end of the discussion; followed by the section on materials and methods.

4. Materials and methods

It would be more customary to put this section before the results, enabling the reader to judge the results on the basis of the methods.

Line 357: ‘promiscuity that does not facilitate application of laws’ – unclear what this means. If there are laws that regulate sale of pharmaceuticals, it would be helpful to briefly describe them, and why, if they are flouted, this occurs.

Line 359: ‘viable community pharmacies were randomly selected’ – unclear what is meant here, or how the community pharmacies were selected. Eg was there a list of pharmacies ? or were pharmacies selected in the field ? If so, how were the geographic areas / locations selected ? Approximately what proportion of all pharmacies in the district were included ?

Line 360 :’availability of their drug sellers’ – seems to suggest that those who agreed to participate were included. What proportion were excluded ? Does this mean that those included are likely to be more compliant with regulation / better practice ?

Line 365: refers to ‘patients who had given their consent’ – the earlier section of the methods did not refer to selection of patients as sources of prescriptions.  How were patients selected ?

This seems to be explained further in line 380

Line 389: refers to classification of prescriptions – but the results are based on classification of the antibiotics prescribed.

Line 399: AWaRe tool not described.

How many pharmacies / drug sellers were approached ? How many patients / public ?

Note line 348 ‘only a few pharmacies’ – how many ?

Line 363 Authorization from the national ethics commission – although extent of ethical review not described; consent was obtained from participants

Line 378 400 prescriptions obtained – basis of selection of this number is not provided; one in 3 patients approached  - but unclear over what period, what was the basis for selection of one in 3 patients, and how the one in three were identified.

Number of pharmacies involved not provided.

Conclusions

While prescribing patterns are important in the emergence of AMR, the study does not provide any information on the extent of AMR in DRC, and hence the potential need for more information on resistance patterns to guide prescribers.

English language is generally fine

Author Response

Thank you for your remarks.

Please find attached below the revised paper as arranged according to the questions you addressed to us.

Also, we would like to mention that according to the point 3.5. Summary, it is better to keep it like it is because it takes into consideration many parameters based on the classification criteria, including continents data and some isolated countries with the references. This is why we find it better to compare using sentences instead of addressing tables, which can lead to major confusions of interpretation.

Best regards,

Jocelyn MANKULU

Reviewer 2 Report

Dear Authors,

This is an interesting piece of research, and adding the AWaRe classification provides a new dimension to the results.

From my perspective, the results are relevant to start designing interventions within the frame of antibiotic stewardship campaigns.

Perhaps three simple points should be highlighted even more in the Abstract and also the Discussion and Conclusions:

(1) 75% of the prescriptions in the sample included at least one antibiotic.

(2) 43% of the prescribed antibiotics were Watch, a proportion higher than Access (36%).

(3) 1 out of 5 antibiotics prescribed in this sample were "not recommended" by WHO.

(4) At least 6% of the prescriptions included three or more antibiotics. Combining antibiotics should be done in exceptional cases under close supervision. It is difficult to justify this in community health.

Finally, I would add a limitation of the study, something like: "Unfortunately, the study design did not allow to know the indications of use (diagnosis)."

And perhaps a comment on "missing" antibiotics. For example, lower urinary tract infections are common conditions. But there is no prescription including nitrofurantoin, one of the recommended Access antibiotics. (WHO AWaRe antibiotic book https://www.who.int/publications/i/item/9789240062382)

Kind regards

Author Response

Thank you for your comments.

Please find attached the article with revised parts highlighted in yellow.

Best regards,

Jocelyn MANKULU

Round 2

Reviewer 1 Report

The majority of the comments and recommendations for revision on the first draft have been satisfactorily addressed. See comments below. 

Introduction:

The rationale in general in regards the importance of AMR is outlined – but could be strengthened with more information on the role / contribution of antibiotic prescribing patterns, and rationale for selection of the district of Tshangu. No information is provided on the characteristics of Tshangu in terms of SES of population, availability / access to medical care, or the number and types of providers of pharmaceuticals.

Revision: More information on the district of Tshangu and the rationale for selection of this district is provided.

Line 136 notes that many medicines are obtained without prescription. A potential limitation of the study is that it focuses on prescriptions and prescribing of antibiotics, and does not capture the potential use of antibiotics without prescription. This could be mentioned.

Revision: line 141 additional information on the use of antibiotics without prescription provided

Methods
It would be customary for the methods section to precede the results, so that the reader could understand how the results were obtained. Suggest that the methods and materials section at the end of the paper in the current draft be moved to this position.

Revision: Methods and materials section moved to follow the introduction as recommended.

Line 357: ‘promiscuity that does not facilitate application of laws’ – unclear what this means. If there are laws that regulate sale of pharmaceuticals, it would be helpful to briefly describe them, and why, if they are flouted, this occurs.

Revision: not addressed

Line 359: ‘viable community pharmacies were randomly selected’ – unclear what is meant here, or how the community pharmacies were selected. Eg was there a list of pharmacies ? or were pharmacies selected in the field ? If so, how were the geographic areas / locations selected ? Approximately what proportion of all pharmacies in the district were included ?

Revision: not addressed

Line 360 :’availability of their drug sellers’ – seems to suggest that those who agreed to participate were included. What proportion were excluded ? Does this mean that those included are likely to be more compliant with regulation / better practice ?

Revision: addressed

Line 365: refers to ‘patients who had given their consent’ – the earlier section of the methods did not refer to selection of patients as sources of prescriptions.  How were patients selected ?

This seems to be explained further in line 380

Line 389: refers to classification of prescriptions – but the results are based on classification of the antibiotics prescribed.

Line 399: AWaRe tool not described.

Revision: addressed (line 176)

How many pharmacies / drug sellers were approached ? How many patients / public ?

Revision: addressed (80 pharmacies)

Note line 348 ‘only a few pharmacies’ – how many ?

Line 363 Authorization from the national ethics commission – although extent of ethical review not described; consent was obtained from participants

Line 378 400 prescriptions obtained – basis of selection of this number is not provided; one in 3 patients approached  - but unclear over what period, what was the basis for selection of one in 3 patients, and how the one in three were identified.

Revision: not addressed

Number of pharmacies involved not provided.

Revision: addressed  (80 pharmacies)

Results

More information on the distribution of the 400 prescriptions would be useful. For example, the number of pharmacies / drug sellers involved (line 348 refers to ‘only a few pharmacies’) and the type of these pharmacies could be relevant to the sample. For example, relatively lower cost pharmacies might attract relatively poorer customers, who might attend the lower cost providers to obtain prescriptions, and thus receive poorer care.

Revision: additional information on pharmacy practices provided

Table 1: The term ‘prescribed one by prescription’ in the Title is not clear. Suggest Classification of antibiotics prescribed as the single agent in prescription.

Revision: Title revised to clarify meaning.

Similarly, the titles for subsequent Tables 2 and 3 could be clarified as : Antibiotics prescribed in prescriptions of two antibiotic agents; and Table 3: Antibiotics prescribed in prescriptions of three antibiotic agents.

Revision: Title revised as recommended

Line 194 Gentamycin – misspelt, should be gentamicin

Revision : line 218 gentamicin correctly spelt; also corrected in Table 3 and subsequently

Discussion

It would be useful to commence the discussion with some comments on the methodology and its impact on the results. In particular the selection of the pharmacies and prescriptions involved and the degree to which these are representative of the use of antibiotics in DRC / Kinshasa.

Revision: Additional information provided in the discussion section on the context of prescribing behaviour in Tshangu.

Line 209-210. The statement that antibiotics were over-used cannot be justified on the basis of information on the number of prescriptions. It would need to be compared with the extent and types  of infectious disease. It might be more appropriate to say that they constitute a large proportion of the prescriptions.

Revision: Addressed to some extent in the discussion and in the conclusions

The discussion section tends to repeat the results while comparing to studies in other countries. It would be appropriate to consider the potential strengths and weaknesses of the study methodology, and factors that might contribute to the high use of watch and not recommended categories, for example the expectation of patients, and existing levels of antibiotic resistance among common pathogens.

Revision: Additional information provided in the conclusions section on limitations of the study and the AWaRE classification

Line 260 ‘medical examination tools’ – this could be further clarified for example ? potential for laboratory examination ? Tendency to rely on ‘clinical’ exam.

Line 266 ‘legal tender in DRC’ – unclear what this means

3.4 Summary

Tends to repeat the information in the results and compare with studies in other countries – but difficult to follow. A table with columns for each study might be easier to compare.

Revision: not addressed

English language is generally satisfactory, with correct usage and grammar. 

Author Response

Revisions Reviewer 1: Second round

Methods:

Line 357: because of the high number of pharmacies in this district that do not comply with the limits set by the country's pharmaceutical legislation authority, which recommends a distance of 500 meters in urban areas and around 1 kilometer in rural areas. And this leads to their concentration where there are high population movements.

Line 359: Targeted pharmacies (of interest) were selected in the field and regarding there frequentation by patients. No geographic limit was set as most of them are concentrated in areas of high population density. However, it should be pointed out that some of the 80 pharmacies we worked on were isolated in certain parts of the district where there were few pharmacies open to the public.

Line 360: This survey only took into account 80 surveyed pharmacies taken randomly without consideration of complying or not. We focused on pharmacies willing to participate and accepting to provide their drug selling strategies and the kind of mostly received prescriptions.

Line 365: See line 380

Line 378: The only criterion for collecting prescriptions was to contain one or more prescribed antibiotics. As patients were not our focus according to this research, we just approached randomly the given few number as an audit parameter without focusing on it as the basic subject of the research, given that our primary target was prescriptions containing antibiotics and, secondarily, the pharmacies where these drugs are dispensed.

Line 260: Due to a lack of appropriate medical equipment (appropriate clinical laboratory) to carry out clinical examinations.

Line 266: These antibiotics are considered to be the antibiotics of choice and are legal to use in the DRC.

3.4. Summary

Table 7. Pooled results from a similar study by continent [25]

Antibiotics

Africa

America

Europe

Southwest Asia

Amoxicillin

6.4%

3.3%

3.6%

2.5%

Amoxicillin + Clav. Acid

4.9%

2.4%

9.1%

8.9%

Azithromycin

2.1%

3.2%%

3.4%

1.9%

Ceftriaxone

12.0%

9.8%

10.1%

15.4%

Ciprofloxacin

3.8%

2.1%

2.9%

2.5%

Gentamicin

10.0%

1.6%

4.3%

0.0%

Metronidazole

5.5%

4.7%

3.7%

5.0%
